# Self-assembled cellulosic superstructures with unanticipated high quantum yields

Cheng Li [1], Zhen Lang[1], Jade Poisson [1], Wenbo Chen [1], Caoxing Huang [1,2] ✉, Evgeny Nimerovsky[3], Philipp Vana [4,5] & Kai Zhang [1,5] ✉

Nonconventional luminophores devoid of traditional, large π-conjugates often suffer from low solid-state fluorescence quantum yields (FLQYs). In parallel, self-assembled bowl-shaped and helical architectures at the micro- and macroscale are unusual (mostly reported at the nanoscale). Here, we report that surface-stearoylated cellulose nanocrystals and cellulose stearoyl esters co-assemble into macroscale helices (FLQY: 86%) with diameters of 32–104 μm. Meanwhile, surface-lauroylated cellulose nanocrystals and cellulose lauroyl esters co-assemble into porous bowl-shaped microparticles (FLQY: 91%) with diameters of 8–19 μm. The high FLQYs are ascribed to the synergism of the dense oxygen clusters and abundant van der Waals interactions and hydrogen bonds between side stearoyl or lauroyl groups, which can promote through-space electron delocalization, ultimately improving fluorescence performance. These results were rationalized by theoretical calculations. Such superstructures exhibit great potential for stable anti-counterfeiting materials due to the excellent regeneration ability as well as structural stability of the oxygen clusters.

In recent years, nonconventional luminophores free of commonly large π-conjugated structures, also called clusteroluminogens, have attracted great attention because of their biocompatibility, environmental friendliness, low cytotoxicity, and flexibility compared to the conventional π-conjugated fluorophores[1–4]. Their emission is proposed to arise from the clustering-triggered emission (CTE) mechanism, wherein aggregation of heteroatoms (e.g., N, O, S) with electron lone pairs induces through-space interaction, facilitating inter- and intramolecular electron delocalization to form cluster-generated chromophores[5–8]. These materials exhibit fascinating photophysical properties, including excitation-dependent luminescence, concentration-enhanced emission, and aggregation-induced emission (AIE)[9,10]. However, they suffer from low quantum yields in the solid state, typically below 10%, which significantly restricts their applications.

Hierarchical architectures constructed via self-assembly processes have garnered significant interest due to their broad applicability in fields ranging from semiconductors[11], energy storage[12], biomaterials[13], sensing[14], optics[15,16], and electronics[17] to catalysis[18]. The self-assembly of these structures is typically driven by weak noncovalent interactions such as host–guest interactions[19], van der Waals interactions[20], hydrogen bonding[21], electrostatic interactions[22,23], hydrophobic interactions[24,25], and π–π stacking[26,27]. Among the diverse morphologies achievable through self-assembly, helical superstructures predominantly manifest at the nanoscale—exemplified by nanofibers[28] and nanoribbons[29]. In contrast, helical architectures with micro- and macroscale dimensions formed via self-assembly remain scarcely reported, underscoring the demand for innovative assembly strategies. Similarly, bowl-shaped superstructures formed via self-

[1]Sustainable Materials and Chemistry, Department of Wood Technology and Wood-based Composites, University of Göttingen, Göttingen, Germany. [2]Co-Innovation Center for Efficient Processing and Utilization of Forest Resources, College of Chemical Engineering, Nanjing Forestry University, Nanjing, China. [3]Department of NMR-Based Structural Biology, Max Planck Institute for Multidisciplinary Sciences, Göttingen, Germany. [4]Institute of Physical Chemistry, University of Göttingen, Göttingen, Germany. [5]Wöhler Research Institute for Sustainable Chemistry (WISCh), University of Göttingen, Göttingen, Germany. ✉e-mail: hcx@njfu.edu.cn; kai.zhang@uni-goettingen.de

assembly are generally confined to the nanoscale and often exhibit smooth, nonporous morphologies[30]. Notably, the fabrication of micro- and macroscale, porous bowl-shaped architectures through self-assembly represents an unmet challenge.

Self-assembly-based structural color and luminescent materials hold significant promise for anti-counterfeiting applications[31–33]. The formability and stability of such materials are critical to their practical applications. On the one hand, the formation of desired self-assembled structures relies on the environmental conditions, which means that the corresponding structural or luminescent colors can be affected by the external conditions in preparation process[34]. On the other hand, the resulting assemblies may not maintain under harsh conditions, since they are often stabilized solely by non-covalent interactions.

Herein, we report the fabrication of highly emissive macroscale helices (MHs) and highly emissive, porous bowl-shaped microparticles (BSMPs) co-assembled from surface-hydrophobized cellulose nano-crystals (CNCs) and cellulose esters, achieving high FLQY values (up to 86% and 91%). First, CNCs with surface-attached stearoyl/lauroyl groups and polymeric cellulose stearoyl/lauroyl esters were allowed to construct the MHs and porous BSMPs via solvent casting method. Then, we investigated the assembled morphologies and fluorescence behavior of six self- and co-assembly systems. To rationalize the high FLQY values observed, theoretical calculations were performed, revealing that the synergism of dense oxygen cluster formation and specific non-covalent interactions are responsible for the highly emissive behavior. Finally, these superstructures were successfully employed to fabricate robust anti-counterfeiting patterns.

## Results

Cellulose stearoyl esters (CSEs) and cellulose lauroyl esters (CLEs) were synthesized via esterification with stearoyl chloride and lauroyl chloride, respectively, at a molar ratio of 2:1 (acyl chloride to hydroxyl groups) (Fig. 1, Supplementary Figs. 1 and 2). Elemental analysis verified a degree of substitution (DS) of 3.0 for both esters, indicating complete functionalization of the cellulose hydroxyl groups. In addition, surface-hydrophobized CNCs, CNC-C18 (surface-attached stearoyl groups) and CNC-C12 (surface-attached lauroyl groups), were prepared via surface esterification using stearoyl chloride and lauroyl chloride, respectively (Figs. 1 and 2a, Supplementary Figs. 1 and 2). The DSs were 0.89 and 0.82 for the CNC-C18 and CNC-C12, as determined by elemental analysis. CNC-C18 has an average length of $75 \pm 25$ nm and an average diameter of $6.2 \pm 2.1$ nm based on the measurements on

100 individual nanorods in the transmission electron microscopy (TEM) image (Fig. 2c). In comparison, CNC-C12 has an average length of $80 \pm 30$ nm and an average diameter of $5.7 \pm 2.4$ nm (Fig. 2d). For CNC-C12 and CNC-C18, the chemical modification of cellulose was regioselective, occurring primarily at C6, followed by C2 and then C3, according to the peak areas in the solid-state NMR spectra (Supplementary Fig. 3, Supplementary Table 1). The zeta potential values of pristine CNC, CNC-C12, and CNC-C18 are $-22.4 \pm 0.4$, $-22.1 \pm 0.7$, and $-21.6 \pm 0.3$ mV, respectively, which indicates that the surface charge can be retained after esterification (Supplementary Fig. 4).

After the self- and co-assembly of surface-hydrophobized CNCs and polymeric cellulose esters, the distinct solid-state FLQYs were achieved via oxygen cluster engineering and tailored non-covalent interactions (Fig. 1). Notably, CSE + CNC-C18 and CLE + CNC-C12 systems gradually co-assembled into two distinct highly emissive architectures: MHs (FLQY: 86%) and porous BSMPs (FLQY: 91%), respectively, during the evaporation of their suspensions in tetra-hydrofuran (THF) on the surface of silicon wafers in the air under ambient conditions (Figs. 1 and 2b–e). The two co-assembly processes are primarily driven by hydrophobic interactions arising from the long aliphatic chains utilized in the functionalization of the CNCs. In addition, a small number of hydrogen bonds, formed by the ester groups and residual hydroxyl groups, also contribute to the assembly.

The CSE + CNC-C18, CSE, CNC-C18, CLE + CNC-C12, CLE, CNC-C12 systems underwent distinct self- and co-assembly pathways, forming structurally divergent superstructures (Supplementary Fig. 5a, b). The MHs co-assembled from CSE + CNC-C18 exhibited diameters of 32–104 µm and consisted of layered nanoplates with thicknesses of 44–94 nm (Fig. 2c, Supplementary Fig. 6). In contrast to the helices co-assembled by CSE and CNC-C18, CSE alone only self-assembled into flower-like microparticles (FLMs) with diameters of 3–7 µm, consisting of nano/meso flaky structures with thicknesses of 70–120 nm (Supplementary Fig. 5a, b). In addition, CNC-C18 alone only self-assembled into nano/mesoscopic rod-like superstructures (18RLSs) with diameters of 50–170 nm and lengths of 120–950 nm. The porous BSMPs co-assembled from CLE + CNC-C12 exhibited diameters of 8–19 µm and heights of 3–15 µm, and were composed of nanosized flaky structures with thicknesses of 70–120 nm (Fig. 2d and Supplementary Fig. 5a, b). In contrast to the BSMPs co-assembled by CLE and CNC-C12, CLE alone only self-assembled into flaky mesostructures (FMs) with thicknesses of 110–390 nm (Supplementary Fig. 5a, b). CNC-C12 alone only self-

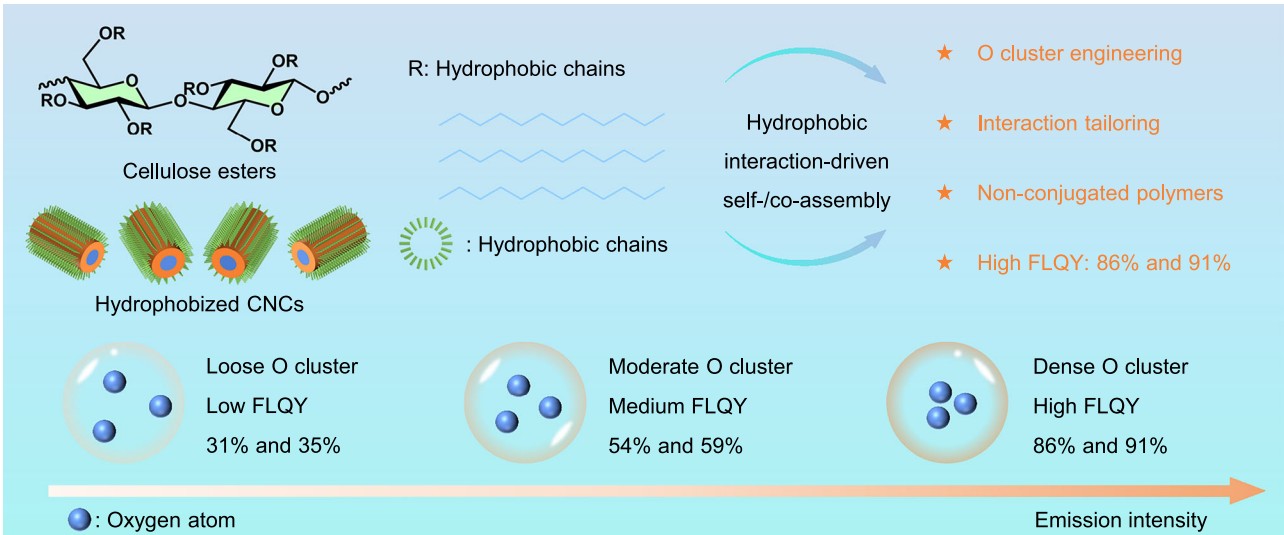

**Fig. 1 | Our strategy to achieve high FLQY.** Schematic illustration of the design strategy for highly emissive nonconjugated superstructures via oxygen cluster engineering and tailored non-covalent interactions.

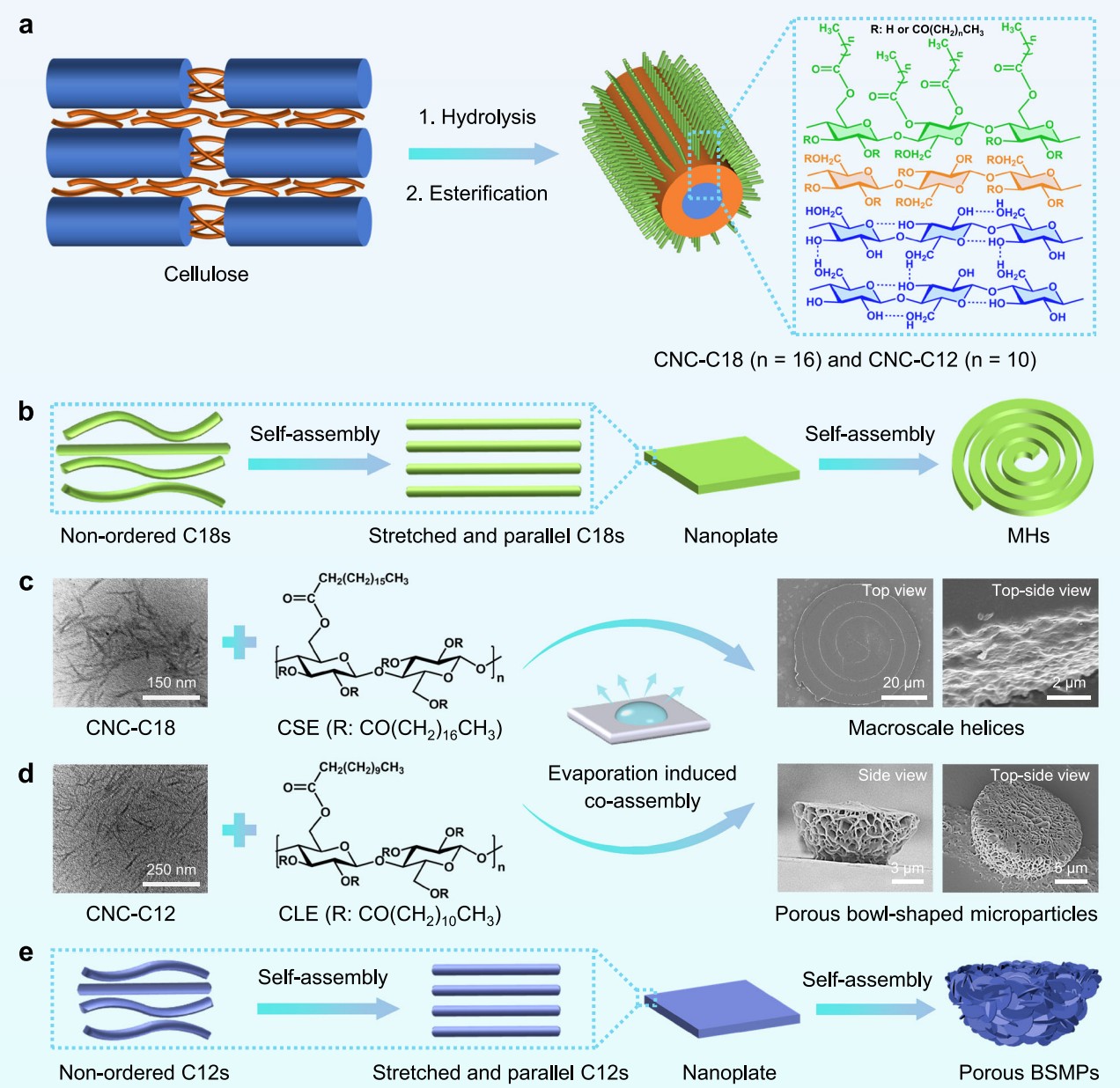

**Fig. 2 | Fabrication of macroscale helices (MHs) and porous bowl-shaped microparticles (BSMPs). a** Schematic illustration of the preparation of CNC-C18 and CNC-C12 via surface functionalization with stearoyl (C18) and lauroyl (C12) groups, respectively. Schematic representation for the assembly process of **b** MHs and **e** porous BSMPs. Formation of the **c** MHs and **d** porous BSMPs driven by evaporation of solvent (tetrahydrofuran), with the insets showing the transmission electron microscopy (TEM) images of CNC-C18/CNC-C12, structural formulas of cellulose esters and scanning electron microscopy (SEM) images of MHs and porous BSMPs.

assembled into nano/mesoscopic rod-like superstructures (12RLSs) with diameters of 45−173 nm and lengths of 135−880 nm.

Molecular dynamics (MD) simulations were conducted to analyze the self- and co-assembly processes of the six above-mentioned systems (Supplementary Fig. 5c, d). To visualize the self- and co-assembly processes, Supplementary Movies 1−6 were derived from simulation trajectories of the later stages approaching the equilibrium states. MD simulations of the two co-assembly systems (CSE + CNC-C18/CLE + CNC-C12) revealed an obvious assembly behavior between the CNC surfaces and surrounding cellulose esters. The trajectories indicate that ester molecules (CSEs/CLEs) are initially attracted to the modified CNC surfaces (CNC-C18/CNC-C12), followed by cooperative self-organization of both components through coupled structural rearrangements, ultimately forming stable conformations (Supplementary

Movies 1 and 4). The conformations of stearoyl/lauroyl groups on the surface-hydrophobized CNCs (CNC-C18/CNC-C12) and cellulose esters (CSEs/CLEs) tend to be stretched and lie parallel in the equilibrium structures of the two co-assembly systems (Supplementary Fig. 5c, d). In comparison, there was a tendency towards stacking of the non-stretched stearoyl and lauroyl groups for the self-assembly of CSE and CLE systems (Supplementary Movies 2 and 5, Supplementary Fig. 5c, d). However, stearoyl and lauroyl groups on adjacent CNC surfaces for self-assembled CNC-C18 and CNC-C12 systems are separated likely due to steric hindrance (Supplementary Movies 3 and 6, Supplementary Fig. 5c, d). Therefore, the hydrophobic interactions between the parallel stearoyl and lauroyl groups are vital to the formation of MHs and BSMPs (Fig. 2b, e and Supplementary Fig. 5c, d). For these six self- and co-assembly systems, variations in the length and conformation of the

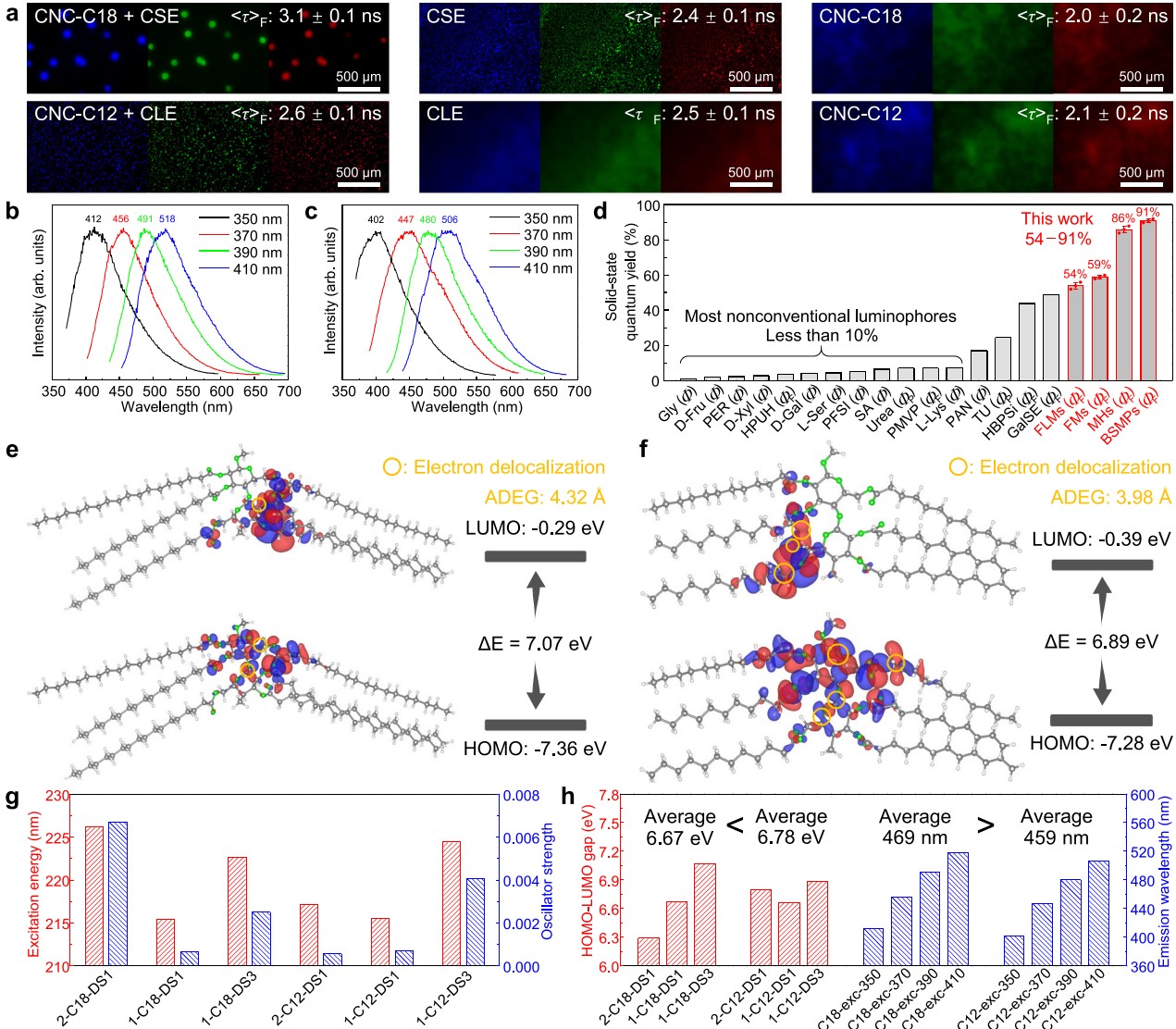

**Fig. 3 | Multicolor fluorescence and density functional theory (DFT) calculations. a** Fluorescence microscopy images of the self- and co-assembled superstructures in three different channels (blue (ex/em) 360/460 nm, green (ex/em) 470/525 nm, red (ex/em) 545/605 nm). The corresponding fluorescence lifetimes are shown on the images ($\lambda_{ex}$ = 350 nm). Prompt emission spectra (normalized) of **b** macroscale helices (MHs) and **c** porous bowl-shaped microparticles (BSMPs) in solid-state at different excitation wavelengths ($\lambda_{ex}$ listed in legend, $\lambda_{max}$ labelled above the peak). **d** Solid-state fluorescence quantum yields of the self/co-assembled superstructures ($\lambda_{ex}$ = 350 nm), in comparison with other reported organic nonconventional luminophores. Gly glycine[50], D-Fru D-fructose[36], PER pentaerythritol[36], D-Xyl D-xylose[36], HPUH hyperbranched polysiloxanes carrying unconjugated carbon−carbon double bonds and hydroxyl groups[51], D-Gal D-galactose[36], L-Ser L-serine[50], PFSI perfluorosulfonate ionomers[52], SA sodium alginate[53], urea[54], PMVP poly(maleic anhydride-alt-vinyl pyrrolidone[55], L-Lys L-lysine[50], PAN polyacrylonitrile[56], TU thiourea[54], HBPSi hyperbranched polysiloxanes[57], GalSE D-galactose stearoyl ester[58]. The calculated highest occupied molecular orbitals (HOMOs) and lowest unoccupied molecular orbitals (LUMOs) of **e** 1-C18-DS3 and **f** 1-C12-DS3. The names indicate the numbers of the repeating unit (1 and 2), side chains (stearoyl (C18) and lauroyl (C12) groups), and the degrees of substitution (1 and 3). The electron delocalization regions are highlighted in the yellow circles, and the average O−O distances for the ester groups (ADEGs) are presented in (**e**, **f**). **g** DFT-calculated excitation energies and oscillator strengths. **h** DFT-calculated HOMO−LUMO energy gaps, correlated with the average emission wavelengths. The names for emission wavelengths indicate the side chains (C18 and C12 groups) and excitation wavelengths (350, 370, 390, 410 nm). Error bars in (**d**) represent means ± SD. Each bar chart column in (**g**, **h**) represents a single measurement/calculation.

hydrophobic chains generate distinct driving forces (hydrophobic interactions), ultimately leading to different morphologies.

Fluorescence microscopy revealed that the six self- and co-assembled superstructures exhibited strong emission in three different channels upon ultraviolet and visible light irradiation (Fig. 3a). Unlike common dyes, whose emission peak positions remain independent of the excitation wavelength, the emission profiles of self-assembled MHs and porous BSMPs are significantly influenced by the excitation wavelength (Fig. 3b, c). For MHs, the emission maxima shifted from 412 to 518 nm as the excitation wavelength increased from

350 to 410 nm. The corresponding full widths at half maximum (FWHMs) of the MHs' spectra at excitation wavelengths of 350, 370, 390, and 410 nm are 88, 84, 89, and 95 nm, respectively (Fig. 3b). Similarly, the emission maxima of BSMPs shifted from 402 to 506 nm as the excitation wavelength increased from 350 to 410 nm. The corresponding FWHMs of the BSMPs' spectra at excitation wavelengths of 350, 370, 390, and 410 nm are 98, 127, 94, and 104 nm, respectively (Fig. 3c). The optimal excitation wavelengths of the excitation spectra of MHs and BSMPs also vary depending on the selected emission wavelength (Supplementary Fig. 7). Moreover, FLMs derived from CSE,

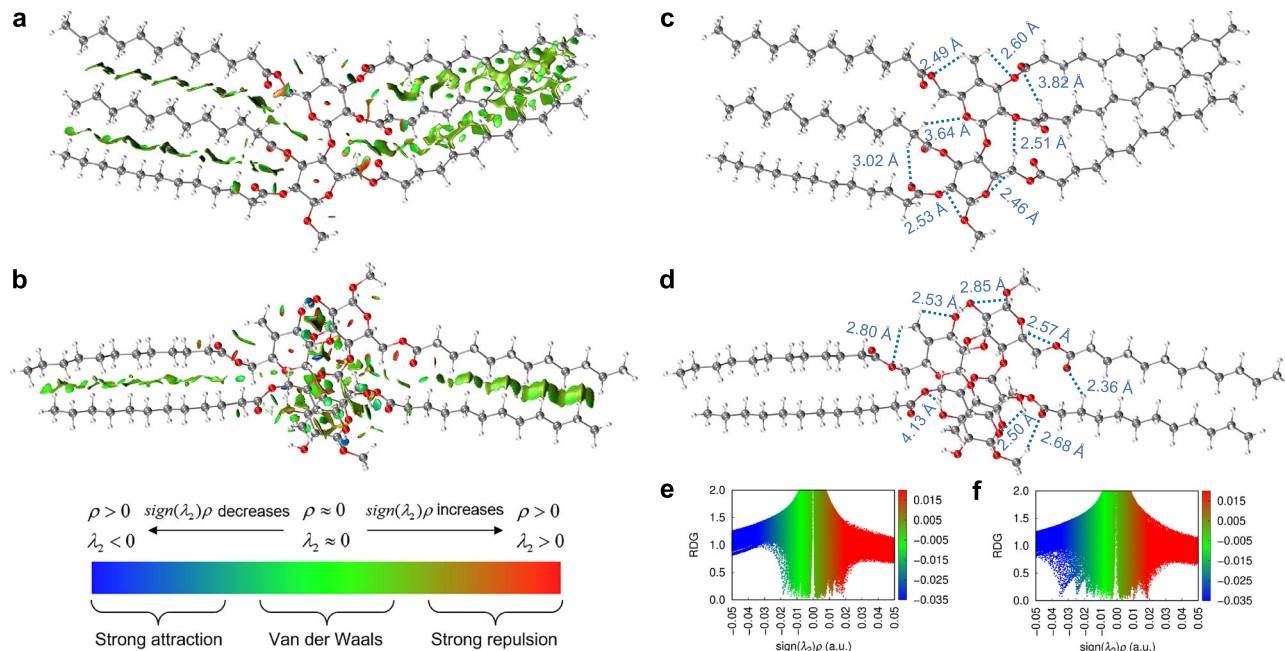

**Fig. 4 | Analysis of the inter- and intramolecular interactions.** Reduced density gradient (RDG) isosurfaces of **a** 1-C12-DS3 and **b** 2-C12-DS1. The names indicate the numbers of the repeating unit (1 and 2), side chains (lauroyl (C12) groups), and the degrees of substitution (1 and 3). Hydrogen bond (C−H···O and O−H···O) distances of **c** 1-C12-DS3 and **d** 2-C12-DS1 in the density functional theory (DFT)-optimized conformations. Scatter plots of RDG versus $sign(\lambda_2)\rho$ for **e** 1-C12-DS3 and **f** 2-C12-DS1.

FMs from CLE, 18RLSs from CNC-C18, and 12RLSs from CNC-C12 also demonstrate the excitation-dependent behavior (Supplementary Fig. 8). The coexistence of diverse emissive clusters with varying degrees of extended electron delocalization is responsible for the excitation-dependent emission. The fluorescence lifetimes of the six self- and co-assembled superstructures are as follows: MHs (CNC-C18 + CSE), 3.1 ± 0.1 ns; FLMs (CSE), 2.4 ± 0.1 ns; 18RLSs (CNC-C18), 2.0 ± 0.2 ns; BSMPs (CNC-C12 + CLE), 2.6 ± 0.1 ns; FMs (CLE), 2.5 ± 0.1 ns; and 12RLSs (CNC-C12), 2.1 ± 0.2 ns.

Remarkably, MHs and porous BSMPs exhibit high solid-state fluorescence quantum yields (FLQYs) of 86% ($\lambda_{ex}$ = 350 nm, $\lambda_{em}$ = 412 nm) and 91% ($\lambda_{ex}$ = 350 nm, $\lambda_{em}$ = 402 nm), respectively (Fig. 3d). Among organic nonconjugated luminophores, only one solid, tri([1,1′-biphenyl]−3-yl)methane (m-TBPM), has been reported to exhibit a higher efficiency (100%)[35]. The performance of MHs and BSMPs starkly contrasts with the majority of such non-conjugated materials, which typically achieve FLQYs below 10% in the solid state (Fig. 3d). In addition, MHs and porous BSMPs exhibit room-temperature phosphorescence (Supplementary Figs. 9 and 10), with very low quantum efficiencies (in contrast to their high FLQYs): $\Phi_P$ (MHs) = 0.7 ± 0.2% ($\lambda_{ex}$ = 350 nm, $\lambda_{em}$ = 485 nm), $\Phi_P$ (BSMPs) = 0.8 ± 0.2% ($\lambda_{ex}$ = 350 nm, $\lambda_{em}$ = 477 nm). FLMs derived from CSE, FMs from CLE, 18RLSs from CNC-C18, and 12RLSs from CNC-C12 demonstrate solid-state FLQYs of 54%, 59%, 31%, and 35%, respectively.

Highest occupied molecular orbitals (HOMOs), lowest unoccupied molecular orbitals (LUMOs), excitation energies and oscillator strengths of one and two repeating units of the cellulose esters with different DS (i.e., 2-C18-DS1, 1-C18-DS1, 1-C18-DS3, 2-C12-DS1, 1-C12-DS1, 1-C12-DS3) were calculated using density functional theory (DFT) (Fig. 3e–h, Supplementary Figs. 11–14). The average HOMO−LUMO energy gap of stearoyl (C18)-modified molecules is lower than that of lauroyl (C12)-modified molecules, consistent with the fluorescence spectra, which show longer excitation and emission wavelengths for the self-assembled CSE + CNC-C18 system compared to the CLE + CNC-C12 system (Fig. 3b, c, h and Supplementary Fig. 7). Furthermore, significant through-space electron delocalization was observed in the

HOMO and LUMO of DFT-optimized conformations for 2-C18-DS1, 1-C18-DS1, 1-C18-DS3, 2-C12-DS1, 1-C12-DS1, and 1-C12-DS3, which could facilitate the excitation and emission processes[9,36]. As shown in the DFT-optimized conformations of 1-C18-DS3 and 1-C12-DS3 (Supplementary Fig. 15), the O−O distances in the cellulose backbone remain nearly constant, whereas those in the side chains (ester groups) vary depending on the substituent chain length. The average O−O distance for the ester groups (ADEGs) of 1-C18-DS3 (4.32 Å) was larger than that observed in 1-C12-DS3 (3.98 Å). Correspondingly, 1-C12-DS3 exhibited enhanced electron cloud overlap and more extensive electron delocalization regions compared to 1-C18-DS3 (Fig. 3e, f), suggesting that compact oxygen clusters can promote through-space electron delocalization.

Since previous studies have shown that increased molecular rigidity and ordering (e.g., crystallinity) can suppress nonradiative decay and enhance CTE[6], we examined the changes in the crystallinity index (CrI) before and after the co-assembly processes. There were two types of crystalline structures that existed in the MHs and porous BSMPs: cellulose (polymorph I) and crystallized fatty-acyl groups. For both crystalline structures, the CrI remains essentially unchanged when comparing co-assembled CLE + CNC-C12 (BSMPs) with pristine CNC-C12 prior to assembly, as well as co-assembled CSE + CNC-C18 (MHs) with pristine CNC-C18 prior to assembly (Supplementary Figs. 16 and 17, Supplementary Table 2). Individual CLE/CSE is not considered, as it is fully dissolved in THF before assembly.

As a class of nonconventional luminophores, the fluorescence of carbohydrates is attributed to abundant oxygen clusters and rationalized by the CTE mechanism in the literature[9]. The weak interactions within clustered chromophores are critical to the luminescence process. Therefore, the inter- and intramolecular non-covalent interactions were analyzed using the reduced density gradient (RDG) method[37]. As shown in Fig. 4a, b, e, f for C12-modified molecules, strong repulsions dominate within the sugar rings, while van der Waals interactions primarily exist between the aliphatic chains. Additionally, the distances of strong attractions, i.e., C−H···O and O−H···O hydrogen bonds, were

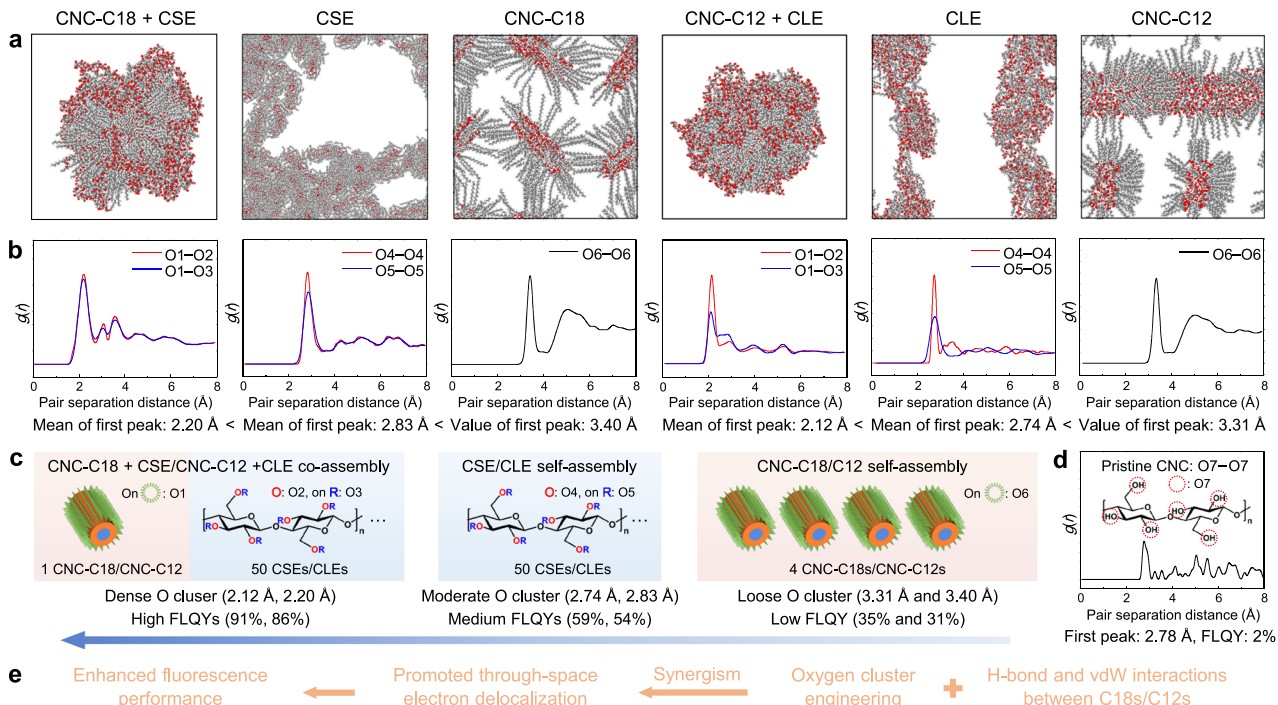

**Fig. 5 | Unveiling the fluorescence mechanism via oxygen cluster engineering and non-covalent interaction tailoring. a** Molecular dynamics (MD)-simulated equilibrium structures and **b** corresponding radial distribution functions (RDFs) of oxygen atoms for the self- and co-assembled CNC-C18 + CSE, CSE, CNC-C18, CNC-C12 + CLE, CLE, and CNC-C12 systems. In panel (**a**), atoms in the snapshots are color-coded by type: carbon in gray, hydrogen in white, and oxygen in red, with the latter highlighted to better visualize oxygen clusters. **c** The numbering scheme for oxygen atoms in (**a, b**) and illustration of the correlation between fluorescence intensity and proximity of oxygen atoms. **d** Calculated RDF for oxygen atoms in pristine CNC (Form I crystalline) without stearoyl (C18) or lauroyl (C12) modifications. **e** The proposed justification for the high fluorescence quantum yields (FLQYs).

measured in the DFT-optimized conformations (Fig. 4c, d). The abundant van der Waals interactions and hydrogen bonds in these clusteroluminogens can contribute to the conformation rigidification, thus diminishing nonradiative decay and favoring fluorescence emission[9].

To rationalize the high FLQYs of MHs and porous BSMPs, radial distribution functions (RDFs) of oxygen atoms were calculated for the six self- and co-assembled systems (CNC-C18 + CSE, CSE, CNC-C18, CNC-C12 + CLE, CLE, and CNC-C12) based on their MD-simulated equilibrium structures (Fig. 5a–c). Only oxygen clusters formed by oxygen atoms in the side chains were analyzed since these clusters are directly linked to the self- and co-assembly processes, with van der Waals interactions and strong attraction between the side chains (stearoyl/lauroyl groups) serving as the driving forces. The position of the first peak in the RDFs corresponds to the shortest O−O distance within the oxygen clusters in the side chains. According to the RDFs, the average value of the first peaks for the O−O distance follows the order: CNC-C18 + CSE (2.20 Å) < CSE (2.83 Å) < CNC-C18 (3.40 Å). Interestingly, the FLQY values are inversely correlated: CNC-C18 + CSE (86%) > CSE (54%) > CNC-C18 (31%). Similarly, for the C12-modified systems, the order of the average value of the first peaks is: CNC-C12 + CLE (2.12 Å) < CLE (2.74 Å) < CNC-C12 (3.31 Å). The FLQY values again follow an opposite trend: CNC-C12 + CLE (91%) > CLE (59%) > CNC-C12 (35%). This inverse correlation is further supported by comparisons between: CNC-C18 + CSE vs. CNC-C12 + CLE (the shortest O−O distance: 2.20 Å > 2.12 Å; FLQY: 86% <91%), CSE vs. CLE (the shortest O−O distance: 2.83 Å > 2.74 Å; FLQY: 54% <59%), and CNC-C18 vs. CNC-C12 (the shortest O−O distance: 3.40 Å > 3.31 Å; FLQY: 31% <35%), all consistently demonstrate that shorter O−O distances correlate with enhanced fluorescence efficiency for these self- and co-assembled clusteroluminogens.

RDF results and FLQY values across the six self- and co-assembled systems reveal dense oxygen clusters as the structural basis for the high FLQY of MHs (CNC-C18 + CSE co-assembly) and porous BSMPs (CNC-C12 + CLE co-assembly). Notably, pristine CNCs without stearoyl or lauroyl modifications exhibit O−O distances (2.78 Å) comparable to those in self-assembled CSE (2.83 Å) and CLE (2.74 Å) systems. However, they achieve only a 2% FLQY (Fig. 5d), which is significantly lower than that of CSE (54%) and CLE (59%). This counterintuitive contrast demonstrates that noncovalent interactions between stearoyl/lauroyl groups are indispensable prerequisites for the observed highly emissive behavior. In addition, the macroscopic morphology of the materials exhibited no effect on their FLQY values (Supplementary Fig. 18). By combining HOMO/LUMO distributions (reflecting varying degrees of through-space electron delocalization) and RDG analyses (identifying non-covalent interaction networks), we conclude that the co-assembled dense oxygen clusters, combined with abundant van der Waals interactions and hydrogen bonding networks between stearoyl/lauroyl groups in MHs and porous BSMPs, synergistically promote conformational rigidification and through-space electron delocalization, ultimately enhancing fluorescence performance (Fig. 5e). Large-scale morphological characteristics do not contribute to the enhancement of fluorescence efficiency (Supplementary Fig. 18).

Anti-counterfeiting patterns were designed using MHs, leveraging their excitation-dependent fluorescence behavior (Fig. 6). The patterns fabricated from the as-prepared MHs appeared white under daylight but displayed distinct fluorescent colors−purple, blue, and yellow−under UV-Vis irradiation at 275, 365, and 495 nm, respectively (Fig. 6a). Importantly, these emission colors and FLQY remained highly stable after various post-

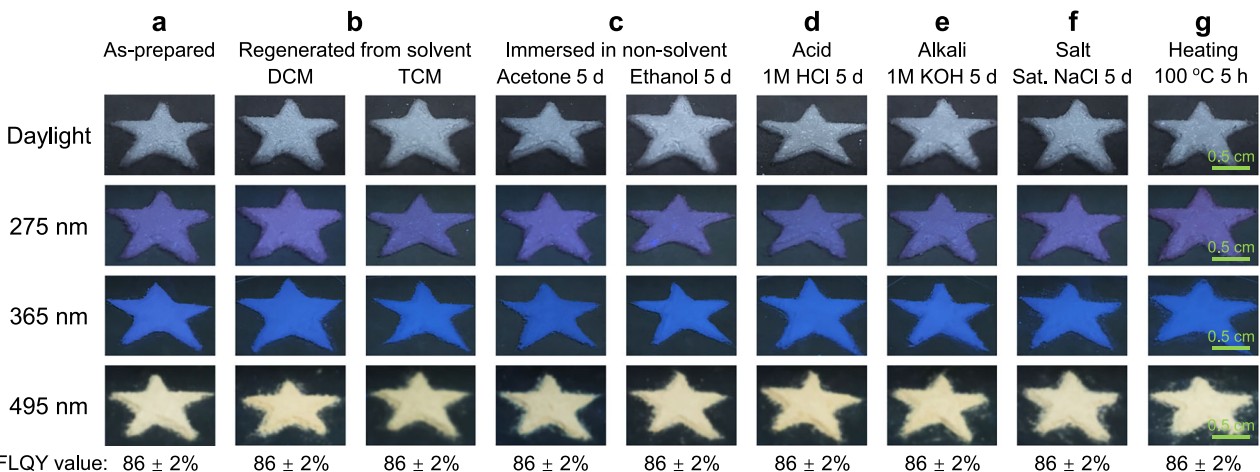

**Fig. 6 | Demonstration of the application of the self-assemblies for stable anti-counterfeiting patterns.** Anti-counterfeiting patterns under daylight and 275, 365, 495 nm UV-Vis irradiation, made by **a** as-prepared MHs, **b** after dissolution and re-assembly in good solvents for CSEs including dichloromethane (DCM) and tri-chloromethane (TCM), **c** after immersion in nonsolvents of CSEs, including acetone and ethanol, for 5 days, **d** after immersion in acidic solution (1 M HCl) for 5 days, **e** after immersion in alkali solution (1 M KOH) for 5 days, **f** after immersion in saturated solution of NaCl for 5 days, **g** after heating at 100 °C for 5 hours. FLQY values ($\lambda_{ex}$ = 350 nm) were shown at the bottom.

treatments (Fig. 6b–g). After dissolution and re-assembly in dichloromethane (DCM) and trichloromethane (TCM), both good solvents for CSEs, the fluorescence colors remained unchanged, indicating that the self-assembled nanoscale oxygen cluster structures are independent of the fabrication method. Further-more, the fluorescence was preserved even after exposure to harsh environments, including immersion in acetone, ethanol, 1 M HCl, 1 M KOH, and saturated solution of NaCl for 5 days, as well as thermal treatment at 100°C for 5 hours (Fig. 6b–g). This stability is attributed to the robust nanoscale organization of oxygen clusters formed during self-assembly. Collectively, these results demonstrate that the oxygen clusters within the superstructures possess excellent regeneration ability and structural stability, underscoring their potential for anti-counterfeiting applications in dynamic or harsh environments.

## Discussion

In summary, we successfully fabricated MHs and porous BSMPs from solutions of cellulose esters and surface-hydrophobized CNCs at room temperature. These superstructures exhibit broad emission profiles and excitation-dependent luminescence. Nota-bly, they also achieve high solid-state FLQY values of 86% and 91%. DFT and MD simulations reveal that the highly emissive behaviors arise from the synergism of the dense oxygen clusters and abundant van der Waals interactions and hydrogen bonds between side stearoyl/lauroyl groups. These superstructures demonstrate strong potential for application in stable anti-counterfeiting materials. Our findings provide inspiration for the design of nonconventional luminophores with both high quantum yield and multicolor emission through oxygen cluster engineering and the tailoring of non-covalent interaction. Mean-while, this work offers valuable insights into the construction of hierarchical architectures through bottom-up self-assembly.

## Methods
### Materials
Commercial microcrystalline cellulose (MCC) with a particle size of 50 µm was purchased from SERVA Electrophoresis GmbH (Heidelberg, Germany). Pyridine, lauroyl chloride (≥98%), and stearoyl chloride (≥90%) were obtained from Sigma-Aldrich Chemie GmbH (Steinheim,

Germany). Deionized water (DI water) was used in all steps, and all other solvents were used directly from TH Geyer (Hamburg, Germany) without further treatment.

### Synthesis of CSE and CLE
CSE and CLE were synthesized under heterogeneous conditions, as previously reported with a few modifications, as indicated below[38].

For synthesis of CSE, MCC (1 g) was pre-dried in a vacuum oven at 60 °C for 3 h to eliminate residual moisture prior to being suspended in 30 mL of pyridine. Then, the cellulose suspension was heated to 100 °C and 13.83 mL of stearoyl chloride was added dropwise to the hot MCC suspension, while the system was purged with nitrogen. After 1 h stirring at 100 °C, the reaction was quenched by pouring the reac-tion mixture into ethanol (200 mL). Subsequently, the precipitate was obtained by centrifugation (7830 rpm) at 4 °C and purified through repeated dissolution in tetrahydrofuran and precipitation in ethanol.

CLE was synthesized analogously to CSE, with modifications in reagent quantities and precipitation solvent: 8.73 mL lauroyl chloride was used as the esterifying agent, and methanol replaced ethanol as the antisolvent for precipitation. All other experimental conditions and purification procedures remained identical to those described for CSE synthesis.

### Preparation of surface-hydrophobized CNC (CNC-C18 and CNC-C12)
Pristine CNCs: CNCs were prepared via HCl hydrolysis of MCC. Spe-cifically, 20 g of MCC was dispersed in 300 mL of 4 mol L$^{-1}$ aqueous HCl solution and stirred for 10 days. The mixture was then heated to 85 °C for 4 hours in the presence of 4 g of FeCl$_3$ as a catalyst. The resulting CNC suspensions were purified and separated via centrifugation. Initially, the suspensions were centrifuged at 2000 rpm, followed by sedimentation at 5000 rpm to obtain uniform CNCs.

CNC-C18: A dispersion of CNCs (30 mg/mL, 30 mL in anhy-drous pyridine) was mechanically stirred under nitrogen atmo-sphere. Stearoyl chloride was added dropwise at a molar ratio of 0.75 relative to the hydroxyl groups on cellulose, and the mixture was heated to 60 °C for 1 h to drive esterification. The reaction mixture was quenched by precipitation into 200 mL of 95% ethanol. The resulting solid was collected via centrifugation (8000 rpm, 10 min), extensively washed with ethanol to remove

residual pyridine and unreacted reagents, and finally redispersed in tetrahydrofuran for subsequent use.

CNC-C12: CNC-C12 was prepared through the same procedure, with lauroyl chloride replacing stearoyl chloride as the acyl donor. All other parameters, including molar ratios, reaction conditions, and post-synthesis purification steps, remained unchanged.

### Self-assembly experiment
Suspensions of CNC-C18 and CNC-C12 (1 mg/mL each) were individually blended with CSE and CLE solutions (1 mg/mL each) at a 1:1 volume ratio. Subsequently, the resulting mixtures (10 μL) were pipetted onto silicon wafers. Superstructures were formed upon solvent (THF) evaporation under ambient conditions. The self-assembly experiments for pristine suspensions of CNC-C18 and CNC-C12, solutions of CSE and CLE (1 mg/mL each) before blending, were carried out using the same procedures.

### Elemental analysis
Elemental analysis was performed on an elemental analyzer Vario EL III CHN instrument from Elementar (Hanau). The degree of substitution (DS) was calculated according to the published method (Supplementary Note 1)[39].

### Zeta potential
Zeta potential of CNC aqueous suspensions (0.2 mg/mL) were measured by a Nano-ZS (Malvern Instruments Ltd., UK). All the samples were tested in triplicate.

### Differential scanning calorimetry (DSC)
DSC measurements were recorded on a NETZSCH/DSC/200/F3/Maia (NETZSCH, Germany) between 20 and 80 °C with a heating rate of 5 K/min. Approximately 10 mg of each sample was used for the measurements, with an empty crucible serving as the reference.

### Attenuated total reflectance (ATR) Fourier-transform infrared (FTIR) spectroscopy
FTIR spectroscopy was conducted on BRUKER ALPHA Spectrometer (Bruker, Germany) at room temperature. All samples were measured between 4000 and 400 cm$^{-1}$ with resolution of 4 cm$^{-1}$ using Platinum ATR.

### Scanning electron microscopy (SEM)
SEM images were captured using a LEO Supra-35 high-resolution field emission scanning electron microscope (Carl Zeiss AG, Germany). A layer of Au/Pd nanoparticles was vacuum-coated on the samples before observation.

### Transmission electron microscopy (TEM)
The surface-hydrophobized CNCs were prepared from their suspensions in THF of about 0.01 wt%. The TEM observation was performed on a CM 12 Transmission Electron Microscope (Philips, Netherland).

### Fluorescence microscopy and spectroscopy
Fluorescence images were acquired with a Keyence BZ-X810 fluorescence microscope at three different excitation/emission wavelengths: blue (ex/em) 360/460 nm, green (ex/em) 470/525 nm, red (ex/em) 545/605 nm. The solid-state excitation and emission spectra, lifetimes, absolute quantum yields were measured on a FLS1000 steady/transient state fluorescence spectrometer equipped with an integrating sphere.

### Solid-state NMR spectroscopy
All experiments were conducted on a Bruker Avance III HD spectrometer operating at 14.1 T (600 MHz $^1$H frequency) using DVT600W2 BL1.3 mm HXY (for samples CNC-C18 and CNC-C12) and 4 mm HXY (for samples CSE, CLE and MCC) probes. The temperature of the nitrogen cooling gas was set to 289 K, with a flow rate of 300–500 liters per hour. For cross polarization transfer from $^1$H to $^{13}$C, a linear ramp of 80% → 100% was applied[40]. Supplementary Table 3 summarizes additional experimental parameters for all five samples, including the magic angle spinning (MAS) rates, recycle delay (D1), total time, cross polarization (CP) conditions, decoupling parameters and acquisition time (AQ).

### Density functional theory (DFT) calculations
All-electron DFT calculations have been carried out by the ORCA quantum chemistry software (Version 6.0.0)[41].

For geometry optimization calculations, B3LYP functional and the def2-SVP basis set[42] were used, and the optimal geometry for each compound was determined (Supplementary Data 1–6). The DFT-D3 dispersion correction with BJ-damping[43,44] was applied to correct the weak interaction to improve the calculation accuracy. The excited states calculations were performed with B3LYP functional and the def2-SVP basis set. Orbital energy level analysis was performed by Multiwfn software[45,46].

The nature of noncovalent interaction was studied by using RDG (reduced density gradient) method[37] through Multiwfn software[45,46]. The visualization of RDG was rendered by VMD[47].

### Molecular dynamics (MD) simulations
Equilibrium molecular dynamics simulation was carried out to investigate the self-assembly at molecular level using LAMMPS[48] under periodic boundary conditions. The surface-hydrophobized CNC (CNC-C18/CNC-C12) was composed of a CNC core and a surface of hydrophobic alkyl chain (stearoyl (C18) and lauroyl (C12) groups), as shown in Supplementary Figs. 19 and 20. The CNC core was modeled using the cellulose-builder developed by Gomes and Skaf at Unicamp[49]. The CNC core is fixed as a rigid body under subsequent simulations, absence of relative movements of atomic positions. The hydrophobic chains were uniformly grafted onto the surface of the CNC, showing parallel arrangement in equilibrium, similar to polymer comb. Note that such conformation is not intentional by modeling, but is a result of densely grafting.

Six groups of simulations were conducted:
(1) Co-assembly of C18-hydrophobized CNC and CSEs: CNC-C18 + CSE.
(2) Self-assembly of CSE.
(3) Self-assembly of CNC-C18.
(4) Co-assembly of C12-hydrophobized CNC and CLEs: CNC-C12 + CLE.
(5) Self-assembly of CLE.
(6) Self-assembly of CNC-C12.

GAFF2 force field and RESP charge were used to model all the macromolecules.

All the systems were relaxed to the equilibrium state at room temperature for at least 1 ns under the NPT ensemble. For better visualization, the volume of the simulated boxes was all set bigger than 180×180×180 Å$^3$. The self-assembly was simulated under NVT ensemble for at least 2 ns. The initial and final configurations were provided in Supplementary Data 7–12. Radial distribution functions (RDFs) were calculated under equilibrium state.

## Data availability
The data generated in this study are provided in the Supplementary Information/Source Data file. All data supporting the findings of this study are available from the corresponding authors upon request. Source data are provided with this paper.

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

## Acknowledgements

K.Z. thanks the EU for financially supporting the project 'NEuM' (ZW7-85191973). C.L. and Z.L. thank the China Scholarship Council (CSC) for the financial support of their PhD grant. J.P. thanks the Alexander von Humboldt Foundation (Ref 3.1-1234770-CAN-HFST-P) and Natural Sciences and Engineering Research Council of Canada (NSERC) (PDF-577833-2023) for postdoctoral fellowships. The authors thank Dr. Florian Ehlers from the Institute of Physical Chemistry, University of Göttingen, for providing device support in fluorescence characterization. The authors thank Dr. Loren B. Andreas from the Department of NMR-Based Structural Biology, Max Planck Institute for Multidisciplinary Sciences, for supporting the NMR measurements. The authors thank Ms. Xintong Meng from the Department of Wood Technology and Wood-based Composites, University of Göttingen, for conducting the XRD measurements used in the responses to the reviewers.

## Author contributions

K.Z. supervised the project. K.Z. and C.L. conceptualized and designed the study. C.L. performed most of the experiments, analyzed the experimental data, wrote the manuscript, and prepared the Supplementary Information. C.H. organized the theoretical calculation. C.H., C.L., and W.C. discussed the simulation. Z.L. prepared the CNCs. J.P. contributed to the revision of the manuscript. P.V. supported the fluorescence spectroscopy measurements. E.N. performed the solid-state NMR measurements. All the authors reviewed the manuscript.

## Funding

## Competing interests

The authors declare no competing interests.
