## [Transparent Peer Review file · Nature Communications]

Self-assembled cellulosic superstructures with unanticipated high quantum yields

Corresponding Author: Professor Kai Zhang

Version 0:

Reviewer comments:

Reviewer #1

(Remarks to the Author)

This is a very good paper that can be accepted with some minor revisions. The work is well-presented and has been carried out in a thorough way. There are a few issues with it that could help improve it in terms of couching the work in the wider context of cellulosic superstructures. For instance, the work discusses supramolecular interactions, which have recently been reviewed <https://www.nature.com/articles/s41578-025-00810-5>. At some points during the discussion it is not clear which supramolecular interactions are key in the formation of the superstructures. I would have thought that with such a large C-chain attached to the sample that this would be hydrophobic, but the authors could be more precise.

Which carbons do the carbon chains during modification attach to? Is it primarily C6, or is it less regioselective? Have the authors carried out any SS NMR to check this? Moreover, are the CNCs that they use charged? If so do the CNCs retain charge or is the modification too high for this to take place? There are examples in the literature of the charge being retained, as well as having hydrophobic groups attached e.g. Nigmatullin et al.

The authors talk about a 'flaky' microstructure, but it is not clear what they mean by this. Is the material brittle, or are they referring to the bowl shaped materials in Figure 2?

What might be the wider applications of these materials, beyond counterfitting? Are there any other possibilities?

Reviewer #2

(Remarks to the Author)

The manuscript reports a novel co-assembly of surface-hydrophobized cellulose nanocrystals (CNCs) with cellulose esters to form macroscale helices (MHs) and porous bowl-shaped microparticles (BSMPs) that exhibit exceptionally high solid-state fluorescence (FL) with quantum yields (FLQY) of 86% and 91%. This is compared to typical nonconjugated luminophores, which usually show very low FLQYs (often <10%, e.g. carboxylated CNC gave only 7.8% in solid). The authors attribute the bright luminescence to dense "oxygen clusters" and extensive van der Waals and hydrogen-bond interactions among the stearyl/lauroyl side chains, promoting through-space electron delocalization (consistent with the clusteroluminescence concept). They also demonstrate broad, excitation-dependent emission colors and suggest potential use in durable anti-counterfeiting materials due to good structural stability. The study combines experimental characterization (morphology, spectroscopy, FLQY measurements) with DFT/MD simulations to elucidate the emission mechanism. Overall, this paper could be accepted once the following issues are addressed.

1) While the manuscript attributes high FLQY to dense oxygen clusters and noncovalent interactions, it remains unclear how crystalline order or amorphous character evolves during the self-assembly process and whether this contributes to fluorescence enhancement. Since previous studies have shown that increased molecular rigidity and ordering (e.g., crystallinity) can suppress nonradiative decay and enhance clusterization-triggered emission (CTE), it is important to evaluate the crystallinity before and after self-assembly.

2) Perform X-ray diffraction (XRD) measurements on the individual components (e.g., CNC-C18, CSE) and the co-assembled MHs and BSMPs. Compare the degree of crystallinity to determine whether assembly induces increased structural order or localized amorphous regions that could facilitate oxygen clustering.

3) If possible, complement with DSC or Raman spectroscopy to support conclusions about molecular packing and rigidity.

4) The authors claim these QYs are “the highest among reported organic nonconjugated luminophores”, However, a very recent work (Wang et al., Nat. Commun. 2024, 15, 6426) reported a nonconjugated organic solid (m-TBPM) with 100% QY. The authors should acknowledge this and either clarify their claim (e.g. highest for cellulose-based systems or for macroscale assemblies) or explain how their results differ (e.g. broader emission, structural features). In any case, citing would update readers and strengthen the discussion. Meanwhile, the authors are strongly encouraged to review these three references (Progress in Polymer Science, 2019, 90, 35-117; Nat. Commun. 2025, 16, 3910; Mater. Today 2020, 32, 275-292) to gain a more comprehensive understanding of this field. It should be noted that the reviewer is not requesting the authors to cite any of these works, but rather offering them as recommended reading to better acquaint themselves with the historical context and current perspectives in this area. I believe the authors have the potential to produce even more interesting and impactful work in the future with these backgrounds.

5) When reporting FLQY, specify the measurement method (integrating sphere, relative method, etc.) and standard used, as these high values require careful calibration.

Reviewer #3

(Remarks to the Author)

This study describes the fabrication of highly emissive macroscale helices (MHs) and porous block-selective microparticles (BSMPs) via co-assembly of surface-hydrophobized cellulose nanocrystals (CNCs) and cellulose esters. This method yields materials with unprecedented quantum yields (up to 86% and 91%) among nonconventional luminophores. The authors present an impressive approach to achieving these superstructures with exceptional efficiencies. I recommend publication with minor revisions to address the points listed below.

1. The first paragraph, for the introduction of the clustering-triggered emission mechanism, other than ref. 4, the following review is highly suitable: Acc. Chem. Res. 2025, 85, 612.
2. The authors attribute the excitation-dependent emission to the “red-edge effect,” but for CTE systems, this behavior typically arises from the coexistence of diverse emissive clusters.
3. MHs and porous BSMPs exhibit exceptional solid-state luminescence quantum yields. Similar high quantum yields were reported for other materials in this work. To ensure accuracy, were these data cross-validated using a different instrument?
4. Normally, nonconventional luminophores like CNCs are phosphorescent. Do these solids exhibit room-temperature or cryotemperature phosphorescence?

Reviewer #4

(Remarks to the Author)

Version 1:

Reviewer comments:

Reviewer #1

(Remarks to the Author)

I am happy with the revisions and the responses to my queries. The paper can now be published.

Reviewer #2

(Remarks to the Author)

I have carefully reviewed the revised manuscript. The authors have thoroughly addressed the issues raised by the reviewers, and I believe the manuscript is suitable for publication in Nature Communications.

Reviewer #3

(Remarks to the Author)

The authors have carefully revised their manuscript to address all the concerns of the reviewers. The current version is acceptable for publication.

Reviewer #4

(Remarks to the Author)

Dear Reviewers,

We very much appreciate your valuable and insightful comments and suggestions. Accordingly, we have conducted supplementary experiments per the recommendations of the reviewers. As well, we revised our manuscript (all changes are shown in red). Included below are the original comments by the Reviewers in black text and our responses in blue.

Reviewer #1 (Remarks to the Author):

This is a very good paper that can be accepted with some minor revisions. The work is well-presented and has been carried out in a thorough way. There are a few issues with it that could help improve it in terms of couching the work in the wider context of cellulosic superstructures. For instance, the work discusses supramolecular interactions, which have recently been reviewed <https://www.nature.com/articles/s41578-025-00810-5>. At some points during the discussion it is not clear which supramolecular interactions are key in the formation of the superstructures. I would have thought that with such a large C-chain attached to the sample that this would be hydrophobic, but the authors could be more precise.

Reply: Thank you for your positive comments and valuable suggestions to improve the quality of our manuscript. We do agree that the hydrophobic interactions are the key driven force for the formation of the superstructures, because of the functionalization with long aliphatic chains. In addition to hydrophobic interactions, we suppose that the small amount of hydrogen bonds, formed by the ester groups as well as the unmodified hydroxyl groups, can also drive the assembly process. Accordingly, we have revised the descriptions in our Revised Manuscript to make this clear. (As we described in our Revised Manuscript, “the two co-assembly processes are primarily driven by hydrophobic interactions arising from the long aliphatic chains utilized in the functionalization of the CNCs. In addition, a small number of hydrogen bonds, formed by the ester groups and residual hydroxyl groups, also contribute to the assembly.”) In addition, we have cited this paper (<https://doi.org/10.1038/s41578-025-00810-5>) mentioned by you in our Revised Manuscript (ref. 19) since it is highly relevant to our

study, in terms of self-assembled cellulosic superstructures.

1) Which carbons do the carbon chains during modification attach to? Is it primarily C6, or is it less regioselective? Have the authors carried out any SS NMR to check this? Moreover, are the CNCs that they use charged? If so do the CNCs retain charge or is the modification too high for this to take place? There are examples in the literature of the charge being retained, as well as having hydrophobic groups attached e.g. Nigmatullin et al.

Reply: We thank the Reviewer for the questions.

For both CSE and CLE, the hydroxyl groups are fully substituted with long aliphatic chains (DS = 3). For CNC-C12 (DS = 0.82) and CNC-C18 (DS = 0.89), the chemical modification of cellulose is regioselective, occurring primarily at C6, followed by C2 and then C3, as indicated by the solid-state NMR spectra (Figure R1 and Table R1). The degree of substitution (DS) for C2 (DS₂) and C3 (DS₃) was determined from the peak splitting (upfield shift) of the adjacent carbons (C1 and C4), whereas the DS for C6 (DS₆) was calculated from the ratio of peaks corresponding to unsubstituted and substituted hydroxyl groups (J. Polym. Sci. A Polym. Chem., 1986, 24, 2981–2993). We have added these results to the Revised Supplementary Information.

Figure R1. Solid-state ¹³C NMR spectra of CNC-C12 showing the peak regions of a)

C1, b) C4, c) C6. Solid-state ^{13}C NMR spectra of CNC-C18 showing the peak regions of d) C1, e) C4, f) C6. $\text{DS}_2 = \text{Area 2} / (\text{Area 1} + \text{Area 2})$, $\text{DS}_3 = \text{Area 5} / (\text{Area 3} + \text{Area 4} + \text{Area 5})$, $\text{DS}_6 = \text{Area 7} / (\text{Area 6} + \text{Area 7} + \text{Area 8})$.

Table R1. Degrees of substitution at C2 (DS_2), C3 (DS_3), and C6 (DS_6) calculated from the solid-state NMR spectra.

Sample	DS_2	DS_3	DS_6
CNC-C12	0.26	0.15	0.39
CNC-C18	0.33	0.10	0.43

The pristine CNCs prior to modification show a zeta potential of -22.4 ± 0.4 mV (Figure R2). They were prepared by hydrochloric acid hydrolysis of MCC and possess only surface hydroxyl groups, which are poorly ionizable, with no readily ionizable groups such as sulfate or carboxyl groups. After esterification with stearyl chloride or lauroyl chloride, the introduced ester groups also exhibit a low degree of ionization. As shown in Figure R2, the surface charge is retained after esterification, with values of -22.1 ± 0.7 mV for CNC-C12 and -21.6 ± 0.3 mV for CNC-C18. We have added these results to the Revised Supplementary Information.

Figure R2. Zeta potential results of a) CNC, b) CNC-C12, and c) CNC-C18.

2) The authors talk about a 'flaky' microstructure, but it is not clear what they mean by this. Is the material brittle, or are they referring to the bowl shaped materials in Figure 2?

Reply: We thank the Reviewer for this question. In addition to the co-assembly system (CNC-C12+CLE), we further analyzed the corresponding self-assembly systems (CNC-C12, CLE), as control experiments, to investigate the effect of initial assembling

units on the oxygen clusters. As we mentioned in our manuscript, ‘in contrast to the BSMPs co-assembled by CLE and CNC-C12, CLE alone only self-assembled into flaky mesostructures (FMs) with thicknesses of 110–390 nm (Figure S7a,b).’ Figure R3 below is a copy of Figure S7 for better review. The ‘flaky’ microstructure refers to the morphology in the fifth column of Figure R3 (Figure S7a,b).

Figure R3. Morphologies of the superstructures and conformations of the hydrophobic chains for the six self- and co-assembly systems (labeled above).

3) What might be the wider applications of these materials, beyond counterfitting? Are there any other possibilities?

Reply: We thank the Reviewer for this comment. Other potential applications include but are not limited to encryption/decryption, molecular imaging, CPL (circularly polarized luminescence) materials, UV absorbers, UV-vis light detection owing to the λ_{ex} -dependent emission.

Reviewer #2 (Remarks to the Author):

The manuscript reports a novel co-assembly of surface-hydrophobized cellulose

nanocrystals (CNCs) with cellulose esters to form macroscale helices (MHs) and porous bowl-shaped microparticles (BSMPs) that exhibit exceptionally high solid-state fluorescence (FL) with quantum yields (FLQY) of 86% and 91%. This is compared to typical nonconjugated luminophores, which usually show very low FLQYs (often <10%, e.g. carboxylated CNC gave only 7.8% in solid). The authors attribute the bright luminescence to dense “oxygen clusters” and extensive van der Waals and hydrogen-bond interactions among the stearyl/lauroyl side chains, promoting through-space electron delocalization (consistent with the clusteroluminescence concept). They also demonstrate broad, excitation-dependent emission colors and suggest potential use in durable anti-counterfeiting materials due to good structural stability. The study combines experimental characterization (morphology, spectroscopy, FLQY measurements) with DFT/MD simulations to elucidate the emission mechanism. Overall, this paper could be accepted once the following issues are addressed.

Reply: We sincerely thank the Reviewer for the constructive input and valuable suggestions, which have been very helpful in improving both the scientific merit and clarity of the discussion of the manuscript.

1) While the manuscript attributes high FLQY to dense oxygen clusters and noncovalent interactions, it remains unclear how crystalline order or amorphous character evolves during the self-assembly process and whether this contributes to fluorescence enhancement. Since previous studies have shown that increased molecular rigidity and ordering (e.g., crystallinity) can suppress nonradiative decay and enhance clusterization-triggered emission (CTE), it is important to evaluate the crystallinity before and after self-assembly.

Reply: We thank the Reviewer for pointing this out and we do agree that it is important to evaluate the crystallinity before and after assembly. The crystallinity changes were evaluated via solid-state NMR, Raman and DSC before and after self-assembly, please find the results of this investigation in the response to Comments 2) and 3) below.

2) Perform X-ray diffraction (XRD) measurements on the individual components (e.g., CNC-C18, CSE) and the co-assembled MHs and BSMPs. Compare the degree of crystallinity to determine whether assembly induces increased structural order or

localized amorphous regions that could facilitate oxygen clustering.

Reply: We thank the Reviewer for this suggestion. For our co-assembly systems, there are two types of crystalline structures: cellulose (polymorph I) and crystallized fatty-acyl groups. We did perform XRD measurements and concluded that this technique is not an appropriate method to determine the crystallinity index (CrI) for either of these crystalline structures (Figure R4). For cellulose (polymorph I), the stacking of long aliphatic chains (stearoyl/lauroyl groups) produces a broad amorphous peak ($2\theta = 16.5\text{--}26.5^\circ$). This peak has also been reported in similar self-assembled systems in the literature (Aggregate 2025, 6, e695; Small 2021, 17, 2102938). As shown in Figure R4, this amorphous peak ($2\theta = 16.5\text{--}26.5^\circ$) overlaps with the characteristic peaks of cellulose. For crystallized fatty-acyl groups, it is particularly challenging to distinguish crystalline and amorphous contributions. On the one hand, their diffraction regions overlap with the characteristic peaks of cellulose (Figure R4). On the other hand, obtaining a single crystal—which would be necessary to resolve the intrinsic crystalline peaks—is difficult. Instead, we employed solid-state NMR and Raman spectroscopy to determine the CrI of cellulose (polymorph I), and DSC measurements to determine the CrI of crystallized fatty-acyl groups (Figures R5–R7, Table R2).

Figure R4. XRD results of the co-assembled BSMPs and MHs.

For cellulose (polymorph I), the CrI determined by solid-state NMR spectroscopy was calculated as the ratio of the crystalline peak area to the total area of the C4 peaks (Chem. Soc. Rev., 2023, 52, 6417–6446; Angew. Chem. Int. Ed. 2020, 59, 3218–3225; Biotechnol Biofuels 3, 10 (2010)) (Figure R5). The results indicate that the CrI remains

essentially unchanged when comparing co-assembled CLE+CNC-C12 (BSMPs) with pristine CNC-C12 prior to assembly, as well as co-assembled CSE+CNC-C18 (MHs) with pristine CNC-C18 prior to assembly. Individual CLE/CSE is not considered, as it is fully dissolved in THF before assembly. We have added the above results to the Revised Manuscript.

Figure R5. Solid-state ^{13}C NMR spectra of a,b) BSMPs co-assembled from CLE+CNC-C12, c) pristine CNC-C12 prior to assembly (partial enlargement of Figure S4b), d,e) MHs co-assembled from CSE+CNC-C18, f) pristine CNC-C18 prior to assembly (partial enlargement of Figure S4a). The stars label the spinning sidebands.

For crystallized fatty-acyl groups, we employed DSC to determine the CrI values (see the response to Comment 3) below).

3) If possible, complement with DSC or Raman spectroscopy to support conclusions about molecular packing and rigidity.

Reply: We thank the Reviewer for this suggestion.

Regarding the CrI for cellulose (polymorph I), we also performed Raman spectroscopy measurements to further support the conclusion (Figure R6). The CrI values were calculated according to the Raman band intensity ratio of the 380 and 1096 cm^{-1}

(Cellulose (2010) 17:721–733). Similarly, the CrI remains essentially unchanged when comparing co-assembled CLE+CNC-C12 (BSMPs) with pristine CNC-C12 prior to assembly, as well as co-assembled CSE+CNC-C18 (MHs) with pristine CNC-C18 prior to assembly. To avoid potential concerns regarding discrepancies in CrI values obtained by different methods, we did not include the Raman spectroscopy results in the Revised Manuscript. Such differences are common, as reported in the literature (e.g., Table 2 in Carbohydrate Polymers 190 (2018) 262–270; Table 2 in Cellulose (2010) 17:721–733).

Figure R6. Raman spectra of a) MHs co-assembled from CSE+CNC-C18, b) pristine CNC-C18 prior to assembly, c) BSMPs co-assembled from CLE+CNC-C12, d) pristine CNC-C12 prior to assembly.

For crystallized fatty-acyl groups, we employed DSC measurements to determine the CrI values (Figure R7 and Table R2). Again, the CrI remains essentially unchanged when comparing co-assembled CSE+CNC-C18 (MHs) with pristine CNC-C18 prior to assembly, as well as co-assembled CLE+CNC-C12 (BSMPs) with pristine CNC-C12 prior to assembly (Figure R7 and Table R2). We have added the above results to the Revised Manuscript.

Figure R7. DSC results of a) MHs co-assembled from CSE+CNC-C18, b) pristine CNC-C18 prior to assembly, c) BSMPs co-assembled from CLE+CNC-C12, d) pristine CNC-C12 prior to assembly.

Table R2. CrI values for crystallized fatty-acyl groups, calculated from the DSC results.

Sample	m_{samp} (mg)	ΔH_f (J/g)	m_{crys} (mg)	DS	m_{acyl} (mg)	CrI
MHs	8.9	29.9	1.204	1.51	6.368	18.9%
CNC-C18	9.9	25.1	1.124	0.89	5.902	19.1%
BSMPs	8.7	29.4	1.157	1.49	5.478	21.1%
CNC-C12	9.5	23.2	0.997	0.82	4.582	21.8%

m_{samp} : mass of the sample used for DSC measurement, ΔH_f : enthalpy of heat fusion, m_{crys} : mass of the crystallized fatty-acyl groups, DS: degree of substitution, m_{acyl} : mass of all fatty-acyl groups, CrI: calculated crystallinity index. For sample MHs, mixing equal masses of CNC-C18 and CSE yields an equivalent DS of 1.51, while for sample BSMPs, mixing equal masses of CNC-C12 and CLE yields an equivalent DS of 1.49. $\text{CrI} = (m_{\text{crys}}/m_{\text{acyl}}) \times 100\%$.

Therefore, in response to the three Comments 1-3), we still attribute the high FLQY to dense oxygen clusters and noncovalent interactions, as originally described. The assembly process induces oxygen cluster formation without enhancing the crystallinity of either cellulose (polymorph I) or crystallized fatty-acyl groups.

4) The authors claim these QYs are “the highest among reported organic nonconjugated luminophores”, However, a very recent work (Wang et al., Nat. Commun. 2024, 15, 6426) reported a nonconjugated organic solid (m-TBPM) with 100% QY. The authors should acknowledge this and either clarify their claim (e.g. highest for cellulose-based systems or for macroscale assemblies) or explain how their results differ (e.g. broader emission, structural features). In any case, citing would update readers and strengthen the discussion. Meanwhile, the authors are strongly encouraged to review these three references (Progress in Polymer Science, 2019, 90, 35-117; Nat. Commun. 2025, 16, 3910; Mater. Today 2020, 32, 275-292) to gain a more comprehensive understanding of this field. It should be noted that the reviewer is not requesting the authors to cite any of these works, but rather offering them as recommended reading to better acquaint themselves with the historical context and current perspectives in this area. I believe the authors have the potential to produce even more interesting and impactful work in the future with these backgrounds.

Reply: We thank the Reviewer for pointing out the imprecise description of the high QYs. Accordingly, we have revised our wording from “the highest among reported organic nonconjugated luminophores” to “the highest among reported biomass-based systems”. In the Revised Manuscript, the following statement was added, “... the highest values reported among biomass-based systems. Within the broader class of organic nonconjugated luminophores, only one solid, m-TBPM, has been reported with a higher efficiency (100%).” We have now included a citation to Wang et al. (Nat. Commun. 2024, 15, 6426), who reported this 100% QY material (ref. 45).

In addition, the three references (Progress in Polymer Science, 2019, 90, 35-117; Nat. Commun. 2025, 16, 3910; Mater. Today 2020, 32, 275-292) comprehensively and systematically discussed the nonconjugated luminophores. These three references, together with the report by Wang et al. (Nat. Commun. 2024, 15, 6426) describing a nonconjugated organic solid (m-TBPM) with 100% QY, are highly relevant to our study and can provide inspiration for readers in this area. Therefore, we have cited all of these references in the Revised Manuscript (ref. 4, 7, 8, 45). We thank the Reviewer again for recommending these papers, which have helped us better understand both the historical context and current perspectives in this field.

5) When reporting FLQY, specify the measurement method (integrating sphere, relative method, etc.) and standard used, as these high values require careful calibration.

Reply: We thank the Reviewer for this comment. Absolute quantum yields were measured on a FLS1000 steady/transient state fluorescence spectrometer equipped with an integrating sphere. We have added this information to the Revised Supplementary Information.

Reviewer #3 (Remarks to the Author):

This study describes the fabrication of highly emissive macroscale helices (MHs) and porous block-selective microparticles (BSMPs) via co-assembly of surface-hydrophobized cellulose nanocrystals (CNCs) and cellulose esters. This method yields materials with unprecedented quantum yields (up to 86% and 91%) among nonconventional luminophores. The authors present an impressive approach to achieving these superstructures with exceptional efficiencies. I recommend publication with minor revisions to address the points listed below.

Reply: We thank the Reviewer very much for evaluating our manuscript, for the positive feedback and for the constructive input. The Reviewer's comments were very valuable to improve the study and its presentation. We have addressed all the Reviewer's suggestions as outlined below.

1. The first paragraph, for the introduction of the clustering-triggered emission mechanism, other than ref. 4, the following review is highly suitable: Acc. Chem. Res. 2025, 85, 612.

Reply: We thank the Reviewer for this suggestion. We were unable to find the reference stated above and have assumed that the Reviewer was referring to Acc. Chem. Res. 2025, 58, 4, 612–624, which has now been added to our Revised Manuscript (ref. 6).

2. The authors attribute the excitation-dependent emission to the “red-edge effect,” but for CTE systems, this behavior typically arises from the coexistence of diverse emissive clusters.

Reply: We thank the Reviewer for pointing this out. We do agree that excitation-dependent behavior typically arises from the coexistence of diverse emissive clusters. Accordingly, we have revised the description in our Revised Manuscript, to attribute the excitation-dependent emission to the coexistence of diverse emissive clusters as well. The following sentence was added to our Revised Manuscript to address this comment, “the coexistence of diverse emissive clusters with varying degrees of extended electron delocalization is responsible for the excitation-dependent emission”.

3. MHs and porous BSMPs exhibit exceptional solid-state luminescence quantum yields. Similar high quantum yields were reported for other materials in this work. To ensure accuracy, were these data cross-validated using a different instrument?

Reply: We thank the Reviewer for this comment. Yes, these data were cross-validated using two FLS1000 spectrometers, with errors of less than 2%. As stated in our manuscript, the FLQYs are $86 \pm 2\%$ and $91 \pm 2\%$, and these values were consistently obtained from both instruments.

4. Normally, nonconventional luminophores like CNCs are phosphorescent. Do these solids exhibit room-temperature or cryotemperature phosphorescence?

Reply: We thank the Reviewer for this question. Yes, these solids exhibit room-temperature phosphorescence (Figures R8 and R9), with very low quantum efficiencies (in contrast to their high FLQYs): Φ_P (MHs) = $0.7 \pm 0.2\%$ ($\lambda_{ex} = 350$ nm, $\lambda_{em} = 485$ nm), Φ_P (BSMPs) = $0.8 \pm 0.2\%$ ($\lambda_{ex} = 350$ nm, $\lambda_{em} = 477$ nm). We have added these phosphorescence results including delayed spectra, lifetimes and quantum efficiencies to our Revised Manuscript.

Figure R8. a) Prompt and b) delayed ($t_d = 0.1$ ms) emission spectra of MHs at different λ_{ex} values at room temperature. c) Prompt and d) delayed ($t_d = 0.1$ ms) emission spectra of BSMPs at different λ_{ex} values at room temperature (λ_{ex} listed in legend, λ_{max} labelled above the peak).

Figure R9. Solid-state phosphorescence lifetimes of a) MHs ($\lambda_{ex} = 350$ nm, $\lambda_{em} = 485$ nm) and b) BSMPs ($\lambda_{ex} = 350$ nm, $\lambda_{em} = 477$ nm) at room temperature.

Reviewer #4 (Remarks to the Author):

I co-reviewed this manuscript with one of the reviewers who provided the listed reports. This is part of the Nature Communications initiative to facilitate training in peer review

and to provide appropriate recognition for Early Career Researchers who co-review manuscripts.

Reply: We greatly appreciate the time and effort you dedicated to providing feedback on our manuscript. The combined reviewer comments are presented and addressed by us in the above text.